# Data Poisoning Won't Save You From Facial Recognition

**Evani Radiya-Dixit** [1]   **Florian Tramèr** [1]

## Abstract

Data poisoning has been proposed as a compelling defense against facial recognition models trained on Web-scraped pictures. By perturbing the images they post online, users can fool models into misclassifying future (unperturbed) pictures.

We demonstrate that this strategy provides a false sense of security, as it ignores an inherent asymmetry between the parties: users' pictures are perturbed *once and for all* before being published and scraped, and must thereafter fool *all future models*—including models trained adaptively against the users' past attacks, or models that use technologies discovered after the attack.

We evaluate two poisoning attacks against large-scale facial recognition, *Fawkes* (500,000+ downloads) and *LowKey*. We demonstrate how an "oblivious" model trainer can simply *wait* for future developments in computer vision to nullify the protection of pictures collected in the past. We further show that an adversary with black-box access to the attack can train a robust model that resists the perturbations of collected pictures.

We caution that facial recognition poisoning will not admit an "arms race" between attackers and defenders. Once perturbed pictures are scraped, the attack cannot be changed so any *future* defense irrevocably undermines users' privacy.

## 1. Introduction

Facial recognition systems pose a serious threat to individual privacy. Various companies routinely scrape the Web for users' pictures to train large-scale facial recognition systems (Hill, Jan; Harwell, 2021), and then make these systems available to law enforcement agencies (Lip-

ton, 2020) or private individuals (Harwell, 2021; Mozur & Krolik, 2019; Wong, 2019).

A growing body of research explores how tools from *adversarial machine learning* can help users fight back (Sharif et al., 2016; Oh et al., 2017; Thys et al., 2019; Shan et al., 2020; Evtimov et al., 2020; Gao et al., 2020; Xu et al., 2020; Yang et al., 2020; Komkov & Petiushko, 2021; Cherepanova et al., 2021; Rajabi et al., 2021; Browne et al., 2020). We revisit a recently proposed approach where users perturb the pictures they post online, in order to *poison* facial recognition models into misidentifying unperturbed pictures (e.g., a picture taken by a stalker or by the police). This idea was popularized by *Fawkes* (Shan et al., 2020), an academic image-perturbation system with 500,000+ downloads, which promises "strong protection against unauthorized [facial recognition] models" (Shan et al., 2021). Following Fawkes' success, other systems have been proposed by academic (Cherepanova et al., 2021; Evtimov et al., 2020) and commercial (Vincent, 2021) parties.

This paper shows that these systems (and, in fact, any poisoning strategy) cannot protect users' privacy. Worse, we argue that these systems offer a false sense of security, as users cannot observe if an attack succeeds or not. Thus, privacy-conscious users might upload perturbed pictures, under the false belief that data poisoning will protect their privacy. Figure 1 shows an overview of our results.

The poisoning attacks deployed by these systems ignore a fundamental asymmetry between Web users and the trainers of facial recognition models. Once a user commits to an attack and uploads a perturbed picture that gets scraped, *this perturbation cannot be changed anymore*. Thus, **if a defense against the user's attack is discovered at any point in the future, the protection offered to the user's past pictures is retroactively lost**. Indeed, facial recognition models are resilient to changes in users' faces over time (Ling et al., 2010). Thus, if the elapsed time before a defense is found is not too high (e.g., less than a decade), the defense can be used to train an accurate model on pictures collected in the past. This holds regardless of any future attacks by the user, since past attacks cannot be changed.

To illustrate this fundamental asymmetry between model trainers and users, we show that a fully "oblivious" model trainer, with no knowledge of users' attacks, can eventually

---

[1]Stanford University. Correspondence to: Evani Radiya-Dixit <evanir@stanford.edu>, Florian Tramèr <tramer@cs.stanford.edu>.

*Accepted by the ICML 2021 workshop on A Blessing in Disguise: The Prospects and Perils of Adversarial Machine Learning.* Copyright 2021 by the author(s).

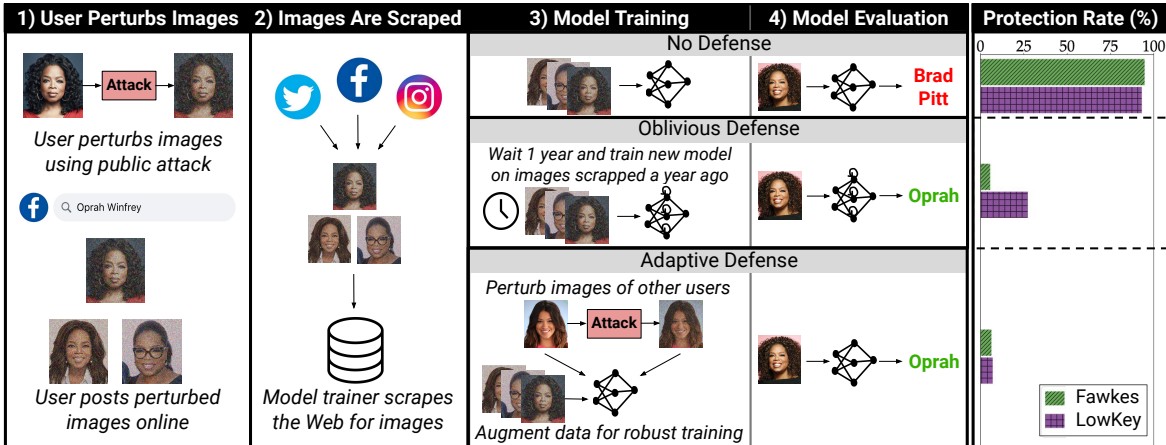

Figure 1: **Attacks and defenses for facial recognition poisoning.** (1) Users perturb their pictures before posting them online. (2) A model trainer continuously scrapes the Web for pictures. (3-4) The model trainer builds a model from collected pictures and evaluates it on *unperturbed* pictures. With no defense strategy, the poisoned model fails to recognize users whose online pictures were perturbed. An "oblivious" model trainer can wait until a better facial recognition model is discovered and retroactively train it on past pictures to resist poisoning. An adaptive model trainer with black-box access to the attack employed by users can immediately train a robust model that resists poisoning.

train an accurate model. The model trainer scrapes pictures from the Web until some progress in facial recognition is made, and then applies the newly discovered method to pictures scraped in the past. For example, we show that attacks produced by Fawkes (released in July 2020) are ineffective if the model trainer switches to a MagFace model (Meng et al., 2021) (released in March 2021). We also find that the more recent LowKey attack (Cherepanova et al., 2021) is only moderately effective against features extracted from OpenAI's CLIP model (Radford et al., 2021).

We further show that if the model trainer knows what attack strategies the users employ, it is not necessary to wait for new progress in facial recognition to nullify the users' attacks. This assumption is realistic for image-perturbation systems that are made *publicly accessible* to cater to a large user base (e.g., Fawkes offers a public tool (Shan et al., 2021), and LowKey offers a Web service (Cherepanova et al.)). We show that with *black-box* access to the perturbation system, an adaptive model trainer can fully circumvent the protections offered by Fawkes and LowKey.

Prior work recognized that model trainers could adapt to attacks (Shan et al., 2021) and predicted an "arms race", where users in turn deploy better attacks. This already happened: Fawkes' latest version counteracts changes in Microsoft's facial recognition system that broke the attack (Shan et al., 2021). Yet, changing the attack is futile, as a model trainer can apply Microsoft's new system to pictures scraped *before* Fawkes updated its attack. Worse, we show that even if the model trainer collects pictures perturbed with Fawkes' new attack, this does not salvage the poor protection of past pictures and the model learns to successfully identify users.

In summary, we argue that poisoning attacks against facial recognition will *not* lead to an "arms race", where new attacks can continuously counteract new defenses. Since the perturbation applied to a picture cannot be changed once the picture is scraped, a successful poisoning attack has to remain effective against *all* future models, even models trained adaptively against the attack, or models that use new techniques discovered only after the attack.

## 2. Data Poisoning for Facial Recognition

### 2.1. Threat Model

We consider a setting where a *user* uploads pictures of themselves to online services such as a social media platform. The user attempts to protect their pictures by adding perturbations that should be almost imperceptible to other people (Szegedy et al., 2013). The user's goal is that a model trained on their perturbed pictures will achieve low accuracy when classifying *unperturbed* pictures of the user. A second party, the *model trainer*, scrapes the Web for pictures to train a large-scale facial recognition model. We assume that the data scraped by the trainer is *labeled*, i.e., all (possibly perturbed) images collected of a user can be assigned to the user's identity.

This setting corresponds to *training-only clean-label* poisoning attacks (Shan et al., 2020; Cherepanova et al., 2021; Evtimov et al., 2020). Keeping with the terminology of the data poisoning literature (Goldblum et al., 2020), we refer to the user as the *attacker* and the trainer as the *defender*.

We provide a more formal treatment of (dynamic) training-only clean-label poisoning attack in Appendix B.

## 2.2. Defenses against Facial Recognition Poisoning

We develop two defense strategies for the model trainer. The first is *oblivious*: the trainer waits until a new type of facial recognition model is developed. The second strategy is *adaptive*: the trainer uses access to the attacker's strategy (as a black-box) to collect perturbed pictures to train on.

**An oblivious defense: time is all you need.** We recall the oblivious defense strategy outlined in Section B:

1. Continuously collect and store labeled poisoned pictures.
2. Wait until a time $t$ where a new type of facial recognition model is invented, preferably one that uses different techniques than prior models.
3. Train that model on the poisoned pictures that were scraped and stored before time $t$, to obtain a classifier $f$.
4. Use the model $f$ for future evaluations, until the next significant progress in facial recognition.

To bypass this defense, a poisoning attack must fool not only today's models, but also all future models.

**An adaptive defense.** Our adaptive defense assumes that the model trainer has black-box access to users' attack. The model trainer collects public data of user faces $\mathbf{X}^{\text{public}}$ (e.g., a canonical dataset of celebrity faces). The model trainer then calls the attack (as a black box), to obtain perturbed samples: $\mathbf{X}^{\text{public}}_{\text{adv}} \leftarrow \texttt{Attack}(\mathbf{X}^{\text{public}})$.

As the model trainer has access to both unperturbed images $\mathbf{X}^{\text{public}}$ and their perturbed versions $\mathbf{X}^{\text{public}}_{\text{adv}}$, they can teach a model to correctly classify both unperturbed and perturbed pictures of these users, and thus encourage the model to learn robust features that generalize to the perturbations applied to other users' pictures.[1]

## 3. Experiments

We evaluate the effectiveness of two facial recognition poisoning tools, Fawkes (Shan et al., 2020) and LowKey (Cherepanova et al., 2021). We show that:

- An oblivious model trainer can collect images perturbed with Fawkes and then use a more recent facial recognition model to retroactively break users' privacy. For the more recent LowKey system, new models can significantly weaken the attack, but not nullify it entirely.
- An adaptive model trainer with black-box access to Fawkes and LowKey can train a robust model that resists poisoning attacks and correctly identifies all users.

---

[1]A black-box adaptive defense might be preventable with an attack that uses *secret per-user randomness* to ensure that robustness to an attack from one user does not generalize to other users. Existing attacks fail to do this, and such an attack would remain vulnerable to our oblivious strategy.

## 3.1. Experimental Setup

We perform all of our experiments with the *FaceScrub* dataset (Ng & Winkler, 2014), which contains over 50,000 images of 530 celebrities. Additional details on the setup for each experiment can be found in Appendix A.

**User configuration.** A user (one of the FaceSrub identities) uses either Fawkes or LowKey to perturb all of their training data (i.e., their pictures uploaded online).

**Model trainer configuration.** The model trainer builds a facial recognition model on all users' labeled training data. We consider three training algorithms:

- *NN:* 1-Nearest Neighbor on top of a feature extractor.
- *Linear:* Linear fine-tuning on top of a frozen feature extractor for 10 epochs.
- *End-to-end:* End-to-end fine-tuning of the feature extractor and linear classifier for 10 epochs.

**Feature extractors.** The model trainer uses a feature extractor trained on VGGFace2 (Cao et al., 2018) and Web-Face (Yi et al., 2014). This extractor is also used in Fawkes v0.3 (Shan et al., 2020). As Fawkes v1.0 and LowKey do not use this extractor, it is as a suitable benchmark for the attacks' transferability. To evaluate oblivious defenses in Section 3.2, we also use three recent feature extractors:

- *Fawkes v1.0:* This is the ArcFace (Deng et al., 2018) feature extractor used in version 1.0 of Fawkes.
- *MagFace:* This is a recent state-of-the-art facial feature extractor (Meng et al., 2021).
- *CLIP:* While not intended for facial recognition, CLIP (Radford et al., 2021) can extract rich facial features (Goh et al., 2021).

**Evaluation Metric.** We evaluate the effectiveness of Fawkes and LowKey by the (top-1) error rate (a.k.a. protection rate) of the facial recognition classifier when evaluated on the *unperturbed* test images of the chosen user. We report the average errors across 20 experiments with a different user in the position of the attacker.

## 3.2. Oblivious Defenses

We begin by evaluating oblivious defenses, where the model trainer waits for a new facial recognition model to be developed and trains it against pictures collected in the past. Figure 2a shows the protection rate of Fawkes against feature extractors over time. The model trainer extracts features from all collected images and trains a NN classifier on top.

If a user had perturbed their pictures using the original Fawkes v0.3 system, they would obtain moderate protection (55% error rate) against a model trained using the same v0.3 feature extractor. However, the same pictures offer no pro-

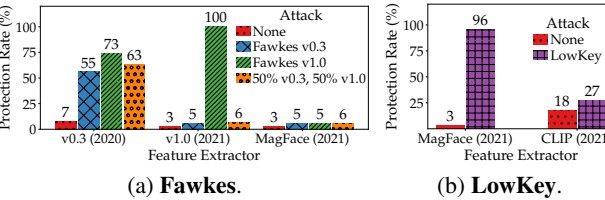

(a) **Fawkes**.      (b) **LowKey**.

Figure 2: **Oblivious defenses against Fawkes and LowKey.** (a) Fawkes version 0.3 is ineffective against the newer feature extractor used in version 1.0 of the tool. Both versions of Fawkes are ineffective against MagFace. (b) LowKey targets today's best models and transfers well to MagFace but not as well to CLIP.

tection against more recent feature extractors ($< 5\%$ error rate). As a result, Fawkes' attack was updated in version 1.0 (Shan et al., 2021) to target a more recent ArcFace (Deng et al., 2018) feature extractor. Yet, while Fawkes v1.0 works well against past and current extractors used by the system ($36–100\%$ error rate), it fails against the recent MagFace extractor (Meng et al., 2021) ($5\%$ error rate).

Moreover, we find that even if the model trainer does use a model that is vulnerable to Fawkes' new attack, users of the old attack cannot "regain" their privacy by adopting the new attack. Specifically, if half a user's pictures were poisoned with Fawkes v0.3, and half are later poisoned with Fawkes v1.0, then a model trainer using the v1.0 extractor (which is vulnerable to the new attack) still defeats the user's attack ($6\%$ protection rate). Thus, once the model trainer has a model that resists past attacks, the protection for pictures perturbed in the past is lost—regardless of future attacks.

LowKey (Cherepanova et al., 2021) fairs somewhat better. LowKey's attack targets an ensemble of state-of-the-art models and transfers well to the recent MagFace model (Meng et al., 2021)—which uses the same training set and architecture as LowKey's extractors (see Figure 2b).

Yet, we predict that LowKey's protections will also fail against models released in the coming years. To motivate this prediction, we show that LowKey only transfers moderately to OpenAI's CLIP (Radford et al., 2021). While CLIP was not trained for facial recognition, it can extract rich facial features (Goh et al., 2021), which yield a moderately low error rate of 18% for a nearest neighbor search on un-poisoned data. Under attack, CLIP's error rate remains below $30\%$. This error rate might already be too low to protect user privacy (e.g., someone could take many pictures of a victim, so that at least one of them gets recognized).

### 3.3. Adaptive Defenses

We have shown that an oblivious defender can wait for progress in facial recognition to break users' poisoning attacks. We next consider a model trainer that does not wish to wait for such progress, and instead adaptively trains a

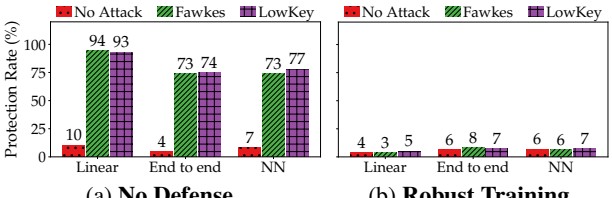

(a) **No Defense.**      (b) **Robust Training.**

Figure 3: **Adaptive defenses against Fawkes and LowKey.** We report (a) the baseline performance (i.e. when no defense is used) for three training modes (Linear, End-to-end, NN); (b) the attack performance after robust training.

model against users' attacks. For all experiments below, the model trainer uses the feature extractor from Fawkes v0.3.

**Baseline.** We first evaluate the baseline performance of the Fawkes (v1.0) and LowKey attacks for the three training strategies that we consider. As shown in Figure 3a, when the model trainer employs no defense strategy, both attacks significantly increase the model's error on the user's unperturbed test images.

**Robust training.** A model trainer with black-box access to the user's poisoning tool can adaptively train a robust facial recognition model as described in Section 2.2. We use the images of half of the FaceScrub users as the public data $\mathbf{X}^{\text{public}}$ that the model trainer feeds to the black-box attack.

When using NNs or linear fine-tuning, we first robustly fine-tune the feature extractor on $\mathbf{X}^{\text{public}}$ and $\mathbf{X}^{\text{public}}_{\text{adv}}$. To evaluate the attack, a NN or linear classifier is trained on top of the robust feature extractor for the entire FaceScrub dataset, including the attacker's perturbed pictures. When the model trainer fine-tunes a model end-to-end, we add $\mathbf{X}^{\text{public}}_{\text{adv}}$ to the model's training set. As shown in Figure 3b, each of the three facial recognition approaches we consider can be made robust. In all cases, the user's protection rate (the test error rate on unperturbed pictures) drops below $8\%$.

## 4. Conclusion

Our work has demonstrated that poisoning attacks cannot save users from large-scale facial recognition models trained on Web-scraped pictures. The initial motivation for these attacks is based on the premise that poisoning attacks can give rise to an "arms race", where better attacks can counteract improved defenses. We have shown that no such arms race can exist, as the model trainer can retroactively apply new models (obtained obliviously or adaptively) to pictures produced by past attacks. To at least counteract an oblivious model trainer, users would have to presume that no significant change will be made to facial recognition models in the coming years. Given the current pace of progress in the field, this assumption is unlikely to hold. Thus, we argue that legislative rather than technological solutions are needed to counteract privacy-invasive facial recognition systems.

## Acknowledgments

We thank Shawn Shan and Emily Wenger for answering our questions about Fawkes, and Micah Goldblum and Valeriia Cherepanova for answering our questions about LowKey. We also thank them, as well as Dan Boneh, Nicholas Carlini, Sanghyun Hong and Alex Kurakin, for helpful discussions and feedback on drafts of this paper.

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

## A. Experimental Details

### A.1. Generation of Perturbed Pictures

For the experiments in Section 3, we generate perturbed images for random FaceScrub (Ng & Winkler, 2014) users using Fawkes (either version 0.3 in "high" mode[2] or the

---

[2] https://github.com/Shawn-Shan/fawkes/tree/63ba2f

most recent version 1.0 in "high" mode[3]) and LowKey.[4]

We use the official pre-aligned and extracted faces from the FaceScrub dataset, and thus disable the automatic face-detection routines in both Fawkes and LowKey. For LowKey, we additionally resize all images to $112 \times 112$ pixels as we found the attack to perform best in this regime.

## A.2. Attack and Model Training Setup

In each of our experiments, we randomly choose one user from the 530 FaceScrub identities to be the attacker. We perturb 100% of the training pictures of that user (70% of *all* pictures) with the chosen attack (Fawkes v0.3, Fawkes v1.0, or LowKey). The training set for the model trainer contains these perturbed pictures, as well as the training pictures of all other 529 FaceScrub users.

For linear fine-tuning and end-to-end fine-tuning, we add a 530-class linear layer on top of a pre-trained feature extractor, and train either only the linear layer, or the entire model. For linear fine-tuning, we train a logistic regression model using `sklearn`. To fine-tune the entire model, we minimize the cross-entropy loss for 500 steps with a batch size of 32 using AdaDelta with learning rate $\eta = 1$. We perform no data augmentation during training.

For nearest neighbor classification, we extract features using a fixed pre-trained feature extractor and assign test points to the same class as the closest training point in feature space.

To report the protection rate conferred by an attack (a.k.a. the model's error rate), we compute the model's error rate on the chosen user's unprotected test pictures. We then average these error rates across 20 experiments, each with a different random attacking user.

## A.3. Oblivious Defenses

To generate Figure 2a, we perturb a user's training pictures using either the Fawkes v0.3 or the Fawkes v1.0 attack, and then use either the Fawkes v0.3 model, the Fawkes v1.0 model or the MagFace model[5] to extract features for a nearest neighbor classifier.

To generate Figure 2b, we perturb a user's training pictures using LowKey, and then use either MagFace or CLIP[6] to extract features for a nearest neighbor classifier.

---

[3] https://github.com/Shawn-Shan/fawkes/tree/5d1c2a
[4] https://openreview.net/forum?id=hJmtwocEqzc
[5] https://github.com/IrvingMeng/MagFace/
[6] https://github.com/openai/CLIP

## A.4. Robust Training

**Data generation using public attacks.** For robust training, we first generate perturbed pictures for many FaceScrub users using different attacks:[7]

Table 1: **Number of FaceScrub users whose images are perturbed for each attack.** Both the perturbed and unperturbed images of these users are used during robust training.

| Attack | Number of users |
|---|---|
| Fawkes v0.3 | 265 |
| Fawkes v1.0 | 50 |
| LowKey | 150 |

Note that this corresponds to 265 users in total (i.e., the users for the Fawkes v1.0 and LowKey attacks are a subset of the users for the Fawkes v0.3 attack). The public dataset $\mathbf{X}^{\text{public}}$ consists of the original pictures of these 265 users, and the perturbed dataset $\mathbf{X}^{\text{public}}_{\text{adv}}$ consists of all the perturbed pictures (across all attacks) of these users.

**Robust model training setup.** For the model trainer, we use the feature extractor from Fawkes v0.3 that the original authors adversarially trained on a dataset different from FaceScrub (Shan et al., 2020).

For linear fine-tuning and nearest neighbors, we first fine-tune this feature extractor on the data from the 265 chosen public users. That is, we add a 265-class linear layer on top of the feature extractor, and fine-tune the entire model end-to-end for 500 steps with batch size 32. To evaluate this robust feature extractor, we pick an attacking user at random (not one of the 265 public users), and build a training set consisting of the perturbed pictures of that user, and the unperturbed pictures of all other 529 users. We then extract features from this training set using the robust model, and fit a linear classifier or nearest neighbor classifier on top.

For end-to-end fine-tuning, we pick an attacking user at random (not one of the 256 public users), and build a training set consisting of: (1) the perturbed pictures of the attacking user; (2) the unperturbed pictures of all other 529 users; (3) the perturbed pictures $\mathbf{X}^{\text{public}}_{\text{adv}}$ of the chosen 265 public users. We then fine-tune the feature extractor with a linear classifier head on this training set for 500 steps with batch size 32.

## A.5. Attack Detection

We also test whether the detector that was trained on one system (i.e., Fawkes or LowKey) transfers to the other (see

---

[7] We started this project by experimenting with Fawkes v0.3 and thus have generated many more perturbed pictures for that attack than for the newer attacks.

Table 2: **Performance of a model trained to detect perturbed images.** Detection performance is very high across all attacks even when smaller perturbations are used (i.e. Fawkes "low" and "mid").

| Attack | Detection Accuracy | Precision | Recall |
|---|---|---|---|
| Fawkes *high* | 99.8% | 99.8% | 99.8% |
| Fawkes *mid* | 99.6% | 99.8% | 99.4% |
| Fawkes *low* | 99.1% | 99.8% | 98.4% |
| LowKey | 99.8% | 99.8% | 99.8% |

Table 3: **Performance of a model trained to detect perturbed images of one attack (source) when evaluated on another attack (destination).**

| Source → Destination | Detection Accuracy | Precision | Recall |
|---|---|---|---|
| Fawkes → LowKey | 99.4% | 99.0% | 99.8% |
| LowKey → Fawkes *high* | 71.9% | 100% | 43.9% |

Table 3).

To evaluate the detectability of perturbed pictures, we choose 45 users and generate perturbations using Fawkes v1.0 (in "low", "mid" and "high" protection modes) and LowKey. We use 25 users during training and 20 users during evaluation. For LowKey, we build a training dataset containing all unperturbed and perturbed pictures of the 25 users. For Fawkes, we do the same but split a user's perturbed pictures equally among the three attack modes.

We then fine-tune a pre-trained MobileNetv2 model (Sandler et al., 2018) on the binary classification task of predicting whether a picture is perturbed. We fine-tune the model for 3 epochs using Adam with learning rate $\eta = 5 \cdot 10^{-5}$. The model is then evaluated by its accuracy on the unperturbed and perturbed pictures of the 20 other users (each user has an equal number of perturbed and unperturbed pictures since we evaluate the Fawkes modes separately).

Table 2 reports detection accuracy and precision and recall scores.

## B. Poisoning Attack Game

We present a standard security game for training-only clean-label poisoning attacks in Figure 4a. We argue that this game fails to properly capture the threat model of our facial recognition scenario.

In this game, the attacker first samples training data $\mathbf{X}, \mathbf{Y}$ from a distribution $\mathbb{D}$ and applies an attack to get the perturbed data $\mathbf{X}_{adv}$. The defender gets the perturbed labeled data $(\mathbf{X}_{adv}, \mathbf{Y})$ and trains a model $f$. The model $f$ is evaluated on *unperturbed* inputs $x$ from the distribution $\mathbb{D}$. For a

given test input $x$, the attacker wins the game if the perturbation of the training data is small (as measured by an oracle $O(\mathbf{X}, \mathbf{X}_{adv}) \mapsto \{0, 1\}$), and if the model misclassifies $x$.

The poisoning game in Figure 4a fails to capture an important facet of the facial recognition problem. The problem is not *static*: users continuously upload new pictures, and the model trainer actively scrapes them to update their model. Below, we introduce a *dynamic* version of the poisoning game, and show how a model trainer can use a *retroactive defense strategy* to win the game. In turn, we discuss how users and model trainers may *adapt* their strategies based on the other party's actions.

**Dynamic poisoning attacks.** To capture the dynamic nature of the facial recognition game, we define a generalized game for clean-label poisoning attacks in Figure 4b. The game now operates in rounds indexed by $i \geq 1$. In each round, the attacker perturbs new pictures and sends them to the defender. The strategies of the attacker and defender may change from one round to the next.

The game in Figure 4b also allows for the data distribution $\mathbb{D}_i$ to change across rounds. Indeed, new users might start uploading pictures, and existing users' faces will change over time. Yet, our thesis is that the main challenge faced by the user (the attacker) is precisely that *the distribution of pictures of their own face changes little over time*. For example, a facial recognition model trained on pictures of a user at 20 years old can recognize pictures of the same user at 30 years old with high accuracy (Ling et al., 2010).

Thus, in each round the defender can reuse training data $(\mathcal{X}_{adv}, \mathcal{Y})$ collected in prior rounds. If the defender scrapes a user's images, the perturbations applied to these images cannot later be changed.

**Retroactive defenses.** The observation above places a very high burden on the attacker. Suppose that in round $i$, the defender discovers a training technique $\texttt{train}_i$ that is resilient to *past* poisoning attacks $\texttt{Attack}_j$ for $j < i$. Then, the defender can train their model solely on the data $(\mathcal{X}_{adv}, \mathcal{Y})$ collected up to round $j$. From there on, the attacker can no longer win the game if the defender ignores future training data (until the defender finds a defense against newer attacks as well). Thus, the attacker's perturbations would need to work against *all future defenses*, even those applied retroactively, for as long as the user's facial features do not naturally change due to age. Note that by design, this retroactive defense does not lead to an "arms race" with future attacks. The defender applies newly discovered defenses to *past* pictures only.

As we will show, this retroactive defense can even be instantiated by a fully *oblivious* model trainer, with no knowledge of users' attacks. The model trainer simply waits for a better

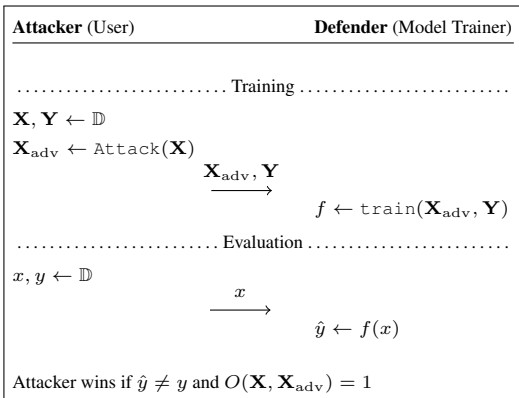

(a) **Game 1: Static game.** The attacker creates a clean-labeled poisoned training set $(\mathbf{X}_{\mathrm{adv}}, \mathbf{Y})$ and the defender trains a model $f$, which is evaluated on *unperturbed* inputs $x$. The attacker wins if $f$ misclassifies $x$ and the poisoned data $\mathbf{X}_{\mathrm{adv}}$ is "close" to the original data $\mathbf{X}$ (according to an oracle $O$).

(b) **Game 2: Dynamic game.** In each round $i \geq 1$, the attacker sends new poisoned data to the defender. The defender may train on *all* the training data $(\boldsymbol{\mathcal{X}}_{\mathrm{adv}}, \boldsymbol{\mathcal{Y}})$ it collected over prior rounds. The strategies of the attacker and defender may change between rounds.

Figure 4: **Security games for training-only clean-label poisoning attacks.**

facial recognition model to be developed, and then applies the model to pictures scraped before the new model was published. This oblivious strategy demonstrates the futility of preventing facial recognition with data poisoning, so long as progress in facial recognition models is expected to continue in the future.

**Adaptive defenses.** A model trainer that does not want to wait for progress in facial recognition can exploit another source of asymmetry over users: *adaptivity*. In our setting, it is easier for the defender to adapt to the attacker, than vice-versa. Indeed, users must perturb their pictures *before* the model trainer scrapes them and feeds them to a secret training algorithm. As the trainer's model $f$ will likely be inaccessible to users, users will have no idea if their attack actually succeeded or not.

In contrast, the users' attack strategy is likely public (at least as a black-box) to support users with minimal technical background. For example, Fawkes offers open-source software to perturb images (Shan et al., 2021), and LowKey (Cherepanova et al.) and DoNotPay (Vincent, 2021) offer a Web API. The defender can thus assemble a dataset of perturbed images and use them to train a model. We call such a defender *adaptive*.

**A note on evasion, backdoor, and obfuscation Attacks.** The security games in Figure 4 assume that the evaluation data is unperturbed. This is the setting considered by Fawkes (Shan et al., 2020) and LowKey (Cherepanova et al., 2021), where a user cannot control the pictures that are fed

to the facial recognition model.

The game dynamics change if the user can use adversarial examples to evade the model (Szegedy et al., 2013; Sharif et al., 2016; Thys et al., 2019; Gao et al., 2020; Cilloni et al., 2020; Rajabi et al., 2021; Oh et al., 2017; Deb et al., 2019; Browne et al., 2020). Such *evasion* attacks favor the attacker: the defender must first commit to a defense and the attacker can then adapt their strategy accordingly (Tramer et al., 2020).

The case of *backdoor* attacks (Chen et al., 2017; Turner et al., 2019; Wenger et al., 2020) is more nuanced. Here, the attacker first poisons the model to react to a specific trigger, and then adds this trigger to inputs at evaluation time. While backdoor attacks involve an attack at evaluation time, this attack is usually *not* adaptive, but merely activates a previously injected trigger. If the initial poisoning attack fails, the attack fails altogether. The inherent advantage of the defender in training-only poisoning attacks thus also applies to backdoor attacks.

Our setting and security game also do not capture *obfuscation* attacks (Newton et al., 2005; Sun et al., 2018a;b; Sam et al., 2020). These attacks either remove or synthetically replace a user's face, and thus fall outside of our threat model.