# OpenReview forum: "Data Poisoning Won’t Save You From Facial Recognition"
_ICML.cc/2021/Workshop/AML — ICML 2021 Workshop AML Oral_

### Official Review · Reviewer_sUm2 · 2021-06-19
**A Very Interesting Paper and Important Findings**

**Rating:** Accept
**Confidence:** 3

**Review:**

In general, this paper argues that poisoning-based methods will not serve as a compelling defense in protecting user privacy of facial images. They argue that this is because the poisoned images will be collected and the adversary can simply wait for future developments or design adaptive methods to nullify the protection of pictures collected in the past.

Pros:
1.	The paper is well-written and the topic is of great significance and is suitable for this workshop.
2.	If no one has studied this aspect so far, then the findings of this paper are novel and very practical for it can reduce user’s false sense of security.
3.	The experiments are also extensive whose results are well support author’s claims.

Cons:
1.	I think the author should provide more details about the adaptive defense in Section 2.2, since this is probably the most technical novel part.
2.	I would suggest the author provide the capacities of both attackers and defenders in Section 2.1 so that the readers can have a better understanding of this `arm-race’.

---

### Decision · Program_Chairs · 2021-06-21

**Decision:**

Accept (Oral)

**Comment:**

This paper argued that poisoning-based methods will not serve as a compelling defense in protecting user privacy of facial images. The paper is interesting and significant for the workshop.